# Using DeepLabCut as a Real-Time and Markerless Tool for Cardiac Physiology Assessment in Zebrafish

**DOI:** 10.3390/biology11081243

**Published:** 2022-08-21

**Authors:** Michael Edbert Suryanto, Ferry Saputra, Kevin Adi Kurnia, Ross D. Vasquez, Marri Jmelou M. Roldan, Kelvin H.-C. Chen, Jong-Chin Huang, Chung-Der Hsiao

**Affiliations:** 1Department of Chemistry, Chung Yuan Christian University, Taoyuan 320314, Taiwan; 2Department of Bioscience Technology, Chung Yuan Christian University, Taoyuan 320314, Taiwan; 3Department of Pharmacy, Research Center for Natural and Applied Sciences, University of Santo Tomas, Manila 1008, Philippines; 4Faculty of Pharmacy, The Graduate School, University of Santo Tomas, Manila 1008, Philippines; 5Department of Applied Chemistry, National Pingtung University, Pingtung 90003, Taiwan; 6Center for Nanotechnology, Chung Yuan Christian University, Taoyuan 320314, Taiwan; 7Research Center for Aquatic Toxicology and Pharmacology, Chung Yuan Christian University, Taoyuan 320314, Taiwan

**Keywords:** DeepLabCut, zebrafish, cardiac physiology, deep learning

## Abstract

**Simple Summary:**

With the advancement of existing technology, artificial intelligence is widely applied in various fields of research, including cardiovascular studies. In this study, we explored the feasibility of conducting a markerless cardiac physiology assessment in zebrafish embryos by using DeepLabCut (DLC), a deep learning tool for motion analysis. Several cardiac parameters, such as heart rate, diastolic–systolic volumes (EDV/ESV), stroke volume, cardiac output, shortening fraction, and ejection fraction were obtained by the DLC-trained model and then compared to the previous published methods, Time Series Analysis and Kymograph. This new method has several advantages, having full automation, precise detection, and real-time labelling. This network was also trained to analyze zebrafish with cardiovascular defects (pericardial edema) induced by chemical treatments with ethanol and ponatinib. It was revealed that the heart rate, EDV/ESV, stroke volume, and cardiac output from both the ethanol and ponatinib groups displayed significant reductions compared with the control. Hopefully, this trained DLC network can contribute to a better understanding and investigation of the existing cardiovascular system and abnormalities.

**Abstract:**

DeepLabCut (DLC) is a deep learning-based tool initially invented for markerless pose estimation in mammals. In this study, we explored the possibility of adopting this tool for conducting markerless cardiac physiology assessment in an important aquatic toxicology model of zebrafish (*Danio rerio*). Initially, high-definition videography was applied to capture heartbeat information at a frame rate of 30 frames per second (fps). Next, 20 videos from different individuals were used to perform convolutional neural network training by labeling the heart chamber (ventricle) with eight landmarks. Using Residual Network (ResNet) 152, a neural network with 152 convolutional neural network layers with 500,000 iterations, we successfully obtained a trained model that can track the heart chamber in a real-time manner. Later, we validated DLC performance with the previously published ImageJ Time Series Analysis (TSA) and Kymograph (KYM) methods. We also evaluated DLC performance by challenging experimental animals with ethanol and ponatinib to induce cardiac abnormality and heartbeat irregularity. The results showed that DLC is more accurate than the TSA method in several parameters tested. The DLC-trained model also detected the ventricle of zebrafish embryos even in the occurrence of heart abnormalities, such as pericardial edema. We believe that this tool is beneficial for research studies, especially for cardiac physiology assessment in zebrafish embryos.

## 1. Introduction

Due to its transparent body and relatively short development processes, the zebrafish is a widely used model for numerous biological studies, including cardiovascular and genetic screenings [1,2]. These fish are also easy to culture, cheap, and have the potential for a high-throughput screening [3]. Zebrafish hearts are typically used for drug testing, which involves the evaluation of different cardiovascular parameters, such as heart rate, the frequency of shortening fraction, stroke volume, ejection fraction, cardiac output, and heartbeat regularity [4]. Numerous methods were developed to detect and quantify the zebrafish heart rate, including visual inspection, electrocardiogram (ECG) use, and image processing methods [5]. However, a visual inspection of the cardiac rate is time-consuming and operator dependent. Using ECG, cardiac events can easily be seen, but recording electrocardiographic traces from embryonic zebrafish requires precisely positioned electrodes, which is an essential step in obtaining reproducible ECG signals [6]. Nowadays, the flexibility to evaluate cardiac performance is mostly based on image processing. By choosing a specific image-processing strategy, the tool can also be used to analyze these animal models. Several previously established methods for cardiac physiology assessment in zebrafish are summarized in Table 1. However, these methods also have limitations which may involve multiple steps before the cardiac function can be obtained.

Another approach is integrating artificial intelligence (AI) deep learning-based methods into cardiovascular research. In recent years, AI and computer vision libraries were used for many applications in the healthcare industry to reduce costs and time, and enhance clinical practices [11]. AI allows accurate information to be produced and large datasets to be quantitatively analyzed, which would not be feasible manually. It integrates omics data with additional layers of information, such as videos, imaging, and other electronic health records [12]. A deep learning-based method required training to compensate for automated analysis of images and demonstrate consistency with the overall assessment [13]. This approach also relies on framework selection to find features in images or object detection [14]. Several deep learning methods established for cardiac physiology assessment in zebrafish are shown in Table 2. However, these methods require some coding skills and advanced computer units. In addition, most of these studies do not provide or share complete details on how the programs were used or run. These deep learning methods also seem to be developing rapidly, making it more difficult to keep track of the details of each procedure.

At present, deep learning has made significant progress in the field of neuroscience, behavior observation, and other analyses related to health examination. Researchers have studied deep learning to train models to track user-defined features for different animals, simplifying traditional feature engineering and image-processing methods [21]. In 2018, Mathis et al. developed an open-source software package called DeepLabCut (DLC), a markerless pose estimation tool to define body parts using deep learning [22]. It combines pre-trained Residual Networks (ResNets) with deconvolutional layers to create a deep convolutional network capable of object recognition and semantic segmentation. This toolbox is provided with several features to extract frames automatically or manually from videos for labeling, create a training dataset based on labeled frames, train networks by selecting the desired framework, and identify the locations of these features in unlabeled data. Basically, DLC enables the creation of tailored part detectors adapted from the body parts of interest that have been labeled. DLC can then be applied to novel videos after a few hours of network training. Although DLC was demonstrated in a mouse [23], fruit fly [24], cat [25], monkey [21], and zebrafish [26] based on the labeling of their outer body parts, there are no inherent limitations. This toolbox can be applied to other models or non-model organisms with a broad range of characteristics.

In this study, we explored the feasibility of conducting a markerless cardiac physiology assessment using DLC in an important aquatic toxicology model, the zebrafish. By demonstrating the practical use of this tool, a better understanding and evaluation of deep learning applications in cardiovascular research can be established. We also demonstrated that trained deep-learning models could generate accurate predictions when given a simple anatomical feature, such as the heart chamber.

## 2. Materials and Methods

### 2.1. Zebrafish Maintenance

Wild-type AB strain zebrafish (*Danio rerio*) were obtained from Taiwan Zebrafish Core Facility (Academia Sinica, Taipei, Taiwan). Sexually matured male and female zebrafish (4 to 6 months old) were used for breeding. An E3 medium and methylene blue mixture were used to culture the zebrafish embryos [27]. The zebrafish embryos were maintained at 25–27 °C with a 14:10 h light:dark cycle. At 2 and 3-days post-fertilization (dpf), zebrafish embryos were used for cardiac physiology assessment. All experiments in this study involving zebrafish were approved by the Institutional Animal Care and Use Committees (IACUCs) of Chung Yuan Christian University (Approval No. 109001, issue date 15 January 2020).

### 2.2. High-Definition Videography

Up to 10 zebrafish were placed onto a microscope slide and mounted with 3% methylcellulose for immobilization. The heart chamber movement (ventricle) for each individual fish embryo was recorded using a high-resolution 4K CCD (XP4K8MA, ToupTek, Zhejiang, China) mounted on an upright microscope (EX20, SOPTOP, Taipei, Taiwan). The video was recorded for 1 min for each individual fish embryo at a resolution of 3840 × 2160 at a rate of 30 fps. The heart position was identical for each recording, and in this case, the head position of animal models was on the left side to maintain consistency throughout the training process. We repeated at least triplicates video recording for microscope slides. 

### 2.3. DeepLabCut Training 

The recorded videos were then used for the training process in DeepLabCut™ (Version 2.2.05, Mathis Group, Swiss Federal Institute of Technology, Lausanne, Switzerland) [22]. The training dataset was generated using a combined 2 and 3-dpf untreated zebrafish (control) group and a chemical treatment (ethanol and ponatinib) group. Ten (10) videos from each group were selected and 20 frames were extracted automatically from each video by OpenCV with a K-means algorithm. In this function, the video was downsampled to frames and clustered based on visual appearance [28]. Generally, this procedure ensures that each extracted frame looks different. After the frames were extracted, each was labeled with eight different points, namely 1, 2, 3, 4, 5, 6, 7, and 8 (Figure 1). The points between 1 and 5 were then connected by a black line to indicate the short axis ($D_{s}$) of the heart chamber. Points 3 and 7 were also connected by a black line to indicate the long axis ($D_{l}$). In this study, as an initial step, we used and compared ResNet-50, 101, and 152 to the image augmentation (imgaug) method to train our labeled data with 50,000 iterations. After the training, the network was evaluated and was ready to be used to analyze the videos. This process was repeated and increased by 50,000 iterations until a total of 500,000 iterations was reached. Finally, based on each ResNet network train-and-test error results, ResNet-152 was selected, due to it having the best performance. The ResNet-152-based neural network [29] with default parameters for 500,000 training iterations was performed. This was validated with ten shuffles, which resulted in the evaluation network of the test-and-train error rate (based on X and Y pixels position). The network was then utilized to analyze new videos and detect the heart chamber of the animal model with similar experimental settings. Afterward, seven cardiac parameters were calculated: end-diastolic volume (EDV), end-systolic volume (ESV), heart rate, SV, CO, EF, and SF.

### 2.4. Cardiac Parameter Calculation

The data calculations were processed in Microsoft^®^ Excel^®^ for Microsoft 365 MSO (Version 2206 Build 16.0.15330.20260) 32-bit. EDV and ESV were calculated with the following equation by assuming the ventricle is spheroid:$$EDV\;or\;ESV=\frac{1}{6}\times \pi \times D_{l}\times D_{s}{}^{2}$$

The data were further processed in Origin 9.1 software (Originlab Corporation, Northampton, MA, USA). By using the Peak Analyzer tool provided in the Origin software, the EDV, ESV, and heart rate were determined. EDV and ESV represent the heart volume during diastolic–systolic phases. Heart rate was defined as the number of times the heart beats per minute (bpm) and was obtained by dividing one minute with the time interval of consecutive beats. After EDV, ESV, and heart rate were obtained, other parameters such as stroke volume, cardiac output, and ejection fraction were measured. Stroke volume represents the volume ejected with every heartbeat (1 cycle) between the EDV and ESV (calculated by the equation below).
Stroke volume=EDV−ESV

The cardiac output is the total amount of blood pumped within a time frame (1 min) and was calculated by the following equation:Cardiac output=Stroke volume×heart rate

The ejection fraction is the volumetric fraction (percentage) change of blood that is pumped out from the heart chamber during diastolic phase which is important to measure heart contractility. It was calculated by the following equation:$$Ejection\;fraction=\frac{Stroke\;volume}{EDV}\times 100\%$$

Lastly, the shortening fraction—another cardiac parameter—represents the percentage reduction in the length of the end-diastolic diameter by the end of systole. Similar to the ejection fraction, it measures the contractility of the heart muscle, and it can be calculated by the following equation:$$Shortening\;fraction=\frac{D_{sd}-D_{ss}}{D_{ss}}\times 100\%$$
where $D_{sd}$ is the short axis diameter during the diastolic stage and $D_{ss}$ refers to the short axis diameter during the systolic stage. All formulations and calculations in this study were conducted based on our previous publications [4,30].

### 2.5. Data Validation with ImageJ and Kymograph

To validate the cardiac parameter results analyzed by DLC, we also applied the ImageJ method with the Time Series Analysis (TSA) plug-in (https://imagej.nih.gov/ij/plugins/time-series.html) (accessed on 22 July 2021) and the kymograph (KYM) generation with broadly applicable routines (BAR) plug-in (https://imagej.net/BAR) (accessed on 8 August 2021) to compare the data. The TSA and KYM methods were performed in ImageJ Version 1.53e on Windows 10 Home. For the TSA method, the peaks of beat intervals were retrieved based on the change in dynamic pixels of blood during diastolic–systolic phases. Meanwhile, in the KYM method, the time-lapse for heart contraction–relaxation images were created in a two-dimensional plot comprising time and space information. Similar output results with total of seven cardiac parameters were calculated. The protocol was conducted based on our previous publications [4,30,31].

### 2.6. Chemical Exposure to Induce Cardiac Abnormality

We further improved our zebrafish cardiovascular detection performance by adding some irregular or abnormal heart morphology images to the training dataset. To achieve this, ethanol and ponatinib were used to induce pericardial edema in zebrafish embryos. Ethanol was purchased from the Sigma-Aldrich Corporation (Taipei, Taiwan), and ponatinib was purchased from Shanghai Aladdin Bio-Chem Technology Co., Ltd. (Shanghai, China). Based on a previous study, 2% ethanol was added to 10-hpf zebrafish [32]. A separate group of 2-dpf zebrafish embryos was exposed to 2.5 ppm of ponatinib for the ethanol-treated group [33]. Pericardial edema was successfully induced in both groups on day 3. All data were recorded using the same method as explained in the previous section.

### 2.7. Heart Rate Variability Measurement by Poincaré Plot

The heart rate variability (HRV) for the control and chemical treatment groups was evaluated using the Poincaré plot plugin (https://www.originlab.com/fileExchange/details.aspx?fid=404) (accessed on 13 March 2022) in Origin 9.1 software (Originlab Corporation, Northampton, MA, USA). The Poincaré plot illustrates HRV through a scatter graph that is useful for quantifying the heart rate regularity or irregularity [34]. The plot was created by inputting the time interval of consecutive heartbeats into the data column. The Poincaré plot was then performed with default settings and with the confidence level for plotting an ellipse: 0.95. The HRV was determined by two indices: the standard deviation 1 (sd1) and standard deviation 2 (sd2) that represented instantaneous and continuous long-term beat intervals, respectively [35].

### 2.8. Statistics

The graphical visualization and statistical analyses were performed using GraphPad Prism software (Version 8.0.2., GraphPad Software, Inc.: San Diego, CA, USA). To compare cardiac physiology using the DLC, TSA, and KYM methods, statistical significance was carried out with repeated measures (RM) one-way ANOVA, followed by the Tukey’s multiple comparisons test. The control and chemical treatment groups were statistically analyzed by ordinary one-way ANOVA, followed by Fisher’s LSD test as the post hoc multiple comparison test. Meanwhile, the sd1 and sd2 of the HRV results were statistically analyzed by the Kruskal–Wallis test, followed by the uncorrected Dunn’s test.

## 3. Results

### 3.1. Overview of Experimental Design

The DeepLabCut (DLC) Python toolbox is versatile, user-friendly, and accessible without needing advanced skills in the programming language. With few trainings of framed images, the network can be trained to achieve human-level labeling accuracy, which makes it applicable for multiple research studies, such as behavior analysis, motion analysis, and medicine. In DLC, the training datasets are prepared based on several initial videos [26]. DLC identifies frames corresponding to the range of movement and object identity based on postures. Once the frames are extracted, the points of interest can be labeled. Checking the accuracy of these annotated frames and correcting them as needed can be done by visual inspection [29]. All extracted labeled frames are then merged and subdivided into train-and-test frames to create the training dataset [22]. To predict the points of interest, a pre-trained network (ResNet) is then refined to adapt its weights according to the labels made by the user. Using the train-and-test frames, one can compare the performance of the trained network. The trained network can be used to analyze videos and extract the pose files [36].

In this study, the cardiac physiology of zebrafish embryos was evaluated by using DLC. The first step was to record the heart movement of this animal model. Since the zebrafish has two heart chambers, it is difficult for the microscope to focus on both chambers. Thus, only one chamber of the ventricle was examined. The video was recorded for 1 min (30 fps) at a 4K resolution of 3840 × 2160 and saved in .mp4 format. After that, 20 frames from each video were automatically extracted using the K-means algorithm. Eight points surrounding the heart chamber were labeled to mark the heart movement. The marker points were then trained using a deep learning approach using a ResNet neural network until 500,000 iterations were reached. The trained datasets were stored in configuration files and used to conduct additional analysis on the novel videos. The heart chamber from the novel videos could then be recognized and labeled automatically (Figure A1). The output data X and Y coordinates were used as a starting point for determining the short axis length, long axis length, and volume (Figure A2), which were used to calculate the cardiac parameters, such as EDV, ESV, stroke volume, heart rate, cardiac output, shortening fraction, and ejection fraction. The experimental design of this study is shown in Figure 1.

### 3.2. DeepLabCut Training for Zebrafish

We adopted the deep neural network, using the ResNet model. This network model has remarkable performance, as it is prepared with information, different capable models, and visual acknowledgment frameworks [37]. The training network was evaluated using different ResNet networks provided in DLC: ResNet-50, 101, and 152. The iteration process started from 50,000 to 500,000, then the train-and-test errors (pixels) were calculated using the p-cutoff method. ResNet-152 provided the best performance on zebrafish compared with ResNet-50 and ResNet-101 (Figure 2). The evaluation results showed that ResNet-101 had the lowest accuracy. This is demonstrated by the results of the train-and-test error rates which were statistically higher than those of ResNet-50 and 152 (Figure 2A). Meanwhile, no significant difference is observed between ResNet-50 and ResNet-152. However, ResNet-152 still performs the best, having the lowest train-and-test error rates. Therefore, the ResNet-152 network was selected for the training process and video analysis in this study. ResNet-152 is a convolutional neural network with 152 layers, and it has features to learn within various abstraction levels to improve its performance [38]. Supporting our data, the ResNet-152 network was reported to achieve the highest accuracy in machine health monitoring. In experiments to predict cancer and detect malignant and benign cells, ResNet-152 performed better than ResNet-18, 50, and 101 [39]. It is an effective recommendation predictor with high training and testing accuracy but requires a longer prediction time for interpretation. 

DLC effectiveness was determined using Euclidean distances between the location labels (x and y coordinates) from training and testing [40]. The location of the centroid of each label was then predicted by ResNet-152. In addition, based on the centroid location, the position of the heart chamber was identified to study its movement. In relation to movement, the x and y coordinates are crucial for counting the distance of pixels to find the short and long axes of the heart chambers. A semi-automatic feature can be achieved by involving machine learning algorithms to complete this task. The trained network provides researchers with a quick and efficient method of quantifying cardiac parameters using animal models studied in a laboratory.

Using ResNet-152, the videos were analyzed, and eight points on the heart chamber outer area were labeled (Figure 3A). After the video was analyzed, the plot position of each point was generated to resemble the trajectory of heart movement. Here, evidence is provided to demonstrate the excellent performance of ResNet-152. Several cases of trajectory positions are displayed in Figure 3. For example, Figure 3B shows the normal condition of heart movement and position without any disturbance during video recording (Appendix A). Meanwhile, the other two figures (Figure 3C,D) display irregular cases with atypical trajectory positions that were generated due to interference caused by a technical problem or by the animal itself. In Figure 3C, the microscope slides moved slightly, but DLC detected and labeled the movement precisely (Appendix A). Another case is shown in Figure 3D, which displays irregular label positions. This incident occurred because the fish suddenly moved during the video recording. However, DLC could still follow the heart chamber position precisely, as shown in the Appendix A.

### 3.3. Cardiac Physiology Comparison between DLC and ImageJ in Control Animals

Data validation was performed for DLC by comparing it to the previously published ImageJ Time Series Analysis (TSA) and kymograph (KYM) methods. Based on the results, the TSA, KYM, and DLC methods did not significantly differ in terms of heart rate (Figure 4A). However, in terms of volume change, we found that there was no significant different between these three methods, except the KYM method displayed a lower ESV compared with DLC and TSA (Figure 4B,C). With the other cardiac parameters, the TSA method displayed significantly more extensive results than DLC, as observed in normalized stroke volume and cardiac output. However, no significant difference was found between DLC and KYM in those parameters (Figure 4D,E). Meanwhile, in two other parameters, shortening fraction and ejection fraction, TSA and KYM displayed significantly higher outcomes than the DLC method (Figure 4F,G). Despite the different outcomes displayed in some cardiac parameters, these methods still succeed in measuring cardiac physiology in the animal model.

### 3.4. Cardiac Physiology Assessment in Zebrafish Embryos after Chemical Treatment

To improve the performance of the DLC network dataset further, we selected two chemicals (ethanol and ponatinib) previously reported to cause cardiovascular defects. Both the ethanol 2% and ponatinib 2.5 ppm groups induced heart deformation and pericardial edema in zebrafish embryos at 3 dpf (Appendix A, respectively). Based on the results, the trained network also detected and labeled the ventricle chamber despite heart malformation due to chemical exposure. After the cardiac parameters were calculated, it was revealed that the heart rate, EDV, ESV, stroke volume, and cardiac output from both the ethanol and ponatinib groups showed significant reduction compared with the control (Figure 5A–E). Meanwhile, for the shortening fraction and ejection fraction endpoints, only the ponatinib group displayed a significant reduction (Figure 5F,G).

Based on the time interval between heartbeats, HRV for the control, ethanol, and ponatinib-treated groups was calculated using the Poincaré plot. Using an ellipse-fitting method, the plot was adjusted using two indices: the standard deviation of instantaneous beat intervals (sd1) and the continuous long-term between two successive peak intervals (sd2). Based on the results, ponatinib caused significant increases in both sd1 and sd2 values, which indicates the higher irregularity of heartbeat compared with the control (Figure 5A,B). In sync with the heartbeat dynamic volume-change patterns (Figure A3), the chemical-treated group, especially ponatinib, had the highest standard deviation value in the Poincaré plot (Figure 6C–E).

## 4. Discussion

In this study, the zebrafish was chosen as an animal model for studying the cardiovascular system. Zebrafish have a rapid reproduction rate, and the embryo has a transparent body, making it possible to observe the heart and its contractility easily. The zebrafish heart has a simple architecture that consists of two major chambers: a ventricle and an atrium [41]. In this study, we focused on ventricular analysis. Throughout the cardiac cycle, the image of the zebrafish heart was imaged in a lateral position with the ventricle section clearly visible, while the atrium was laid outside the plane of focus. Compared with the atrium, the ventricle is primarily spherical and elongated (ovoid) in morphology, with a spherical index of 0.83 [42]. On the other hand, the atrium has an irregular shape which is ineffective for morphology-based quantification. In clinical analysis, ventricular length is more reliable because it can also be measured during echocardiography [43]. Several publications also preferred to focus only on the ventricular area of the zebrafish for cardiovascular research [44,45,46].

### 4.1. Advantages and Limitations

The main advantage of the DLC code is that it provides a graphical user interface (GUI) with simple steps for frame extraction, point labeling, training, and video analysis to retrieve the pose data [22]. Without putting any visible markers on the locations of interest, the system can achieve human-level accuracy using only a small number of training images. Furthermore, DLC is a free, open-source software, with a large-scale discussion forum. It is an excellent motion analysis tool applied in diverse organisms, such as rodents, primates, insects, and fish. The deep features allow DLC to extract body parts despite various background challenges or camera distortions [47]. In addition, DLC possesses a refinement step to take advantage of different scenarios for improving tracking performance [48]. DLC can predict the points without requiring consistency across the frames. It can also retrieve or locate the points of some features that are not initially detected due to occlusions or motion blurs, a capacity lacking in other tracking methods [49,50].

A limitation of DLC is that this software requires modern computational hardware, such as graphical processor units (GPUs), in order to deliver fast and efficient results. GPUs are necessary to manage and improve memory to accelerate graphics rendering, which is useful for machine learning [51]. Another consideration is that the convolutional networks analyze images based on the scale in pixel size; thus, larger images will be processed more slowly [52]. In this study, we considered using 1280 × 720 as maximum resolution videos; however, the choice of resolution still depends on the performance of each computer. Furthermore, DLC is designed for general purpose, so it cannot track any occluded points and does not rely on heuristics, such as a body model [53]. In addition, high-resolution videos were required in this study to observe and monitor heart movement. Thus, we used a high-quality charged-coupled device (CCD) and digital microscopy with higher magnification to capture the heart chamber. With this set-up, DLC delivered sharp and detailed visual colors allowing us to clearly mark and label the edge of the heart. Camera settings were also adjusted to ensure optimal recording results.

### 4.2. Cardiac Physiology Comparison between DLC and ImageJ Methods

In comparison, these three methods displayed identical results in terms of cardiac rhythm, as indicated by the heart rate values. In the TSA method, the heartbeat was measured by dynamic pixel changes of blood in the heart chamber. The dynamic pixel changes showed that, during the systolic phase, the pixel intensity increased when the hemocytes were pumped out from the heart chamber, while during the diastolic phase, lower pixel intensity was observed [31]. On the contrary, DLC measured the heart volume changes by the location of labels generated. Meanwhile, the KYM method used a similar approach by detecting heart chamber movement (contraction and relaxation), but was still limited by manual selection. Each method in this study successfully measured the heart rate equally well, which demonstrates that these methods are able to identify and determine the diastolic–systolic stage (heartbeats) either by pixel intensity changes or chamber movement. However, the statistical comparison showed significant differences in some cardiac parameters between the newly developed method (DLC) and the previously published methods (TSA and KYM). To obtain the cardiac parameters from the TSA and KYM methods, it required several steps, including manual counting [5]. In addition, most of the tools in the ImageJ software is a freehand ruler, which might cause inaccuracy, especially if the heart chambers are small [15]. Since the regions of interest (ROI) in the TSA method are selected manually by using a circle tool with a limited size selection [31], the results might be affected depending on the users. The heart diameter at the diastolic stage (heart relaxation) and systolic stage (heart contraction) were then manually measured by drawing a straight line on a single frame as representatives in ImageJ. Similarly, for the KYM method, the user must draw a line from the inner part of the heart to the outermost area during diastole. This line is fixed; thus, the heart movement detection is limited by the predefined line boundaries. In addition, ROI selection is also limited due to possible image noise (random variation of brightness) potential in certain heart regions. Another problem is that fish embryos might float in the mounting medium, which can cause a shift in the pixel intensity pattern. However, in DLC, the shift or floating movement from the animal models in the mounting medium can still be analyzed without any problems. Unlike the TSA and KYM methods, DLC also provides functionality that enables users to confirm the output results based on the labeled videos and x–y positions of whole video frames. This evidence might explain the difference in cardiac performance results retrieved from these two different methods since the outcomes of the ImageJ-based method is mainly affected by the ROI selection. The limitations of the TSA and KYM methods are also the reason for researchers using a different approach, such as the machine learning method, to collect and process data more comprehensively. Based on these findings, we conclude that the DLC method is more advanced and reliable because it uses a machine learning approach, requires less user involvement, and includes functionality that enables users to confirm the reliability of the training model.

### 4.3. Comparison of the Cardiac Parameters between Control and Chemical-Treated Zebrafish

As previously reported, ethanol and ponatinib exposure causes heart developmental defects [33,54]. Li et al., 2016 reported that zebrafish larvae exposed to 2% ethanol displayed incomplete and damaged blood vessels, alteration in permeability, decreased blood volume, and a deformed heart [32]. Similar to our study, the heart volume became much smaller, and the heart rate decreased. Pericardial edema was observed, caused by the damaged dorsal aorta and arteries that disrupted the blood and fluid circulation [55]. Ponatinib, as low as 3 µM, was reported to induce severe cardiac edema, blood vessel disorders, and narrowing of the dorsal aorta [33]. Similar to our study, ponatinib was also reported to reduce the shortening and ejection fraction in zebrafish [56]. In terms of heart rate variability, ponatinib induces high irregularity. This chemotherapeutic agent is known to be highly related to arrhythmias and contributes to various ECG changes [57]. All the cardiovascular disorders in the experiment were well evaluated using the newly developed method. Despite the abnormalities caused by ethanol and ponatinib, our DLC-training network performed well in defining the ventricle chamber of zebrafish embryos. Shrinkage and downsizing of the atrium chamber were also observed in both chemical-treated groups. The present results also provide additional evidence that it is preferable to focus the analysis on the ventricular section since its morphology is still well recognizable despite alteration of cardiac morphology due to chemical exposure.

## 5. Conclusions

In conclusion, a trained DLC model for automatic detection of the ventricle chamber in zebrafish was established. Using the ResNet-152 network, we retrieved the x–y coordinates of each labeled position, which were then used to calculate several multiple cardiac parameters. This study also revealed that the DLC method displayed identical results in term of cardiac rhythm compared to the previous published TSA and KYM methods. However, in some cardiac parameters, different results were obtained, which might be due to manual measurement and ROI selection-dependency from the ImageJ-based method. Compared with these previous methods, DLC has the advantage of real-time labeling of the whole video frames, with full automation. In addition, the DLC model was also trained to recognize the ventricle chamber despite disruption in cardiac morphology and pericardial edema resulting from ethanol or ponatinib exposure. With this trained model, improvement and increased robustness in detection in zebrafish embryo heart videos were achieved. In the future, hopefully, this trained DLC network can be enhanced with additional training dataset videos from various cardiovascular studies so that this model network can further contribute to a better understanding and investigation of the existing cardiovascular system and abnormalities. It is also necessary to compare this newly developed method to another deep learning tool that uses a similar approach in the future.

## Figures and Tables

**Figure 1 biology-11-01243-f001:**
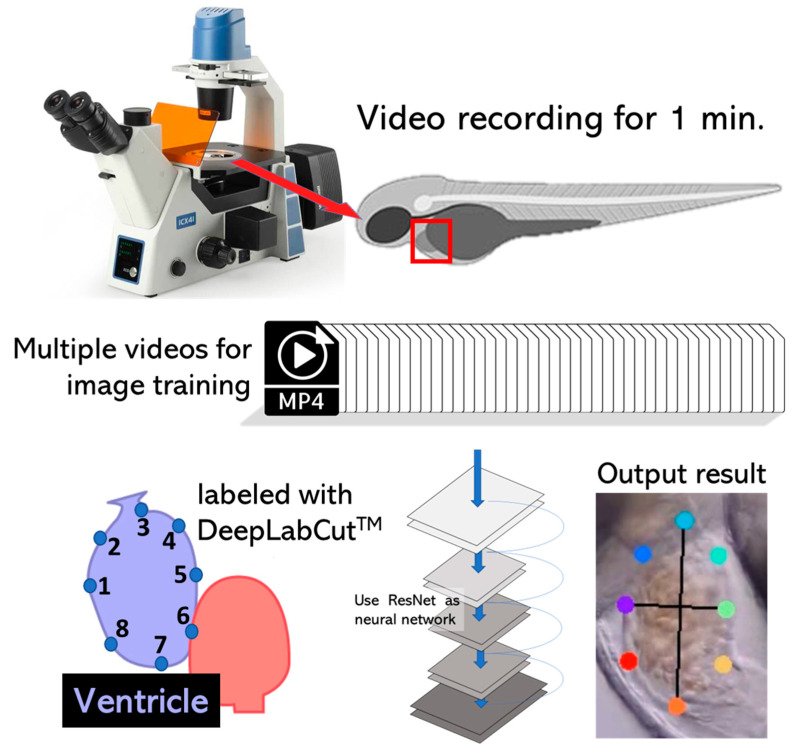
The experimental design was used to detect and label the heart chamber of zebrafish in this study. On top, the animal model: zebrafish (*Danio rerio*) was used in this study as it is widely used for toxicity studies since its body transparency makes it suitable for cardiovascular assessment. Up to 20 videos of a heart beating with a duration of 1 min were collected. The bottom section describes how DLC performed the training process for dataset and video analysis, resulting in a labeled zebrafish ventricle heart chamber.

**Figure 2 biology-11-01243-f002:**
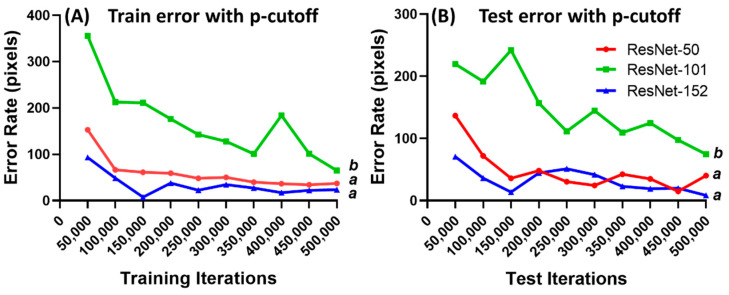
Train error rate (**A**) and test error rate (**B**) with p-cutoff evaluation results with 500,000 iterations in zebrafish heart chambers. The statistical difference was analyzed using ordinary one-way ANOVA, followed by Dunn’s multiple comparison test. The different letters (a and b) indicate significant differences with *p* < 0.05.

**Figure 3 biology-11-01243-f003:**
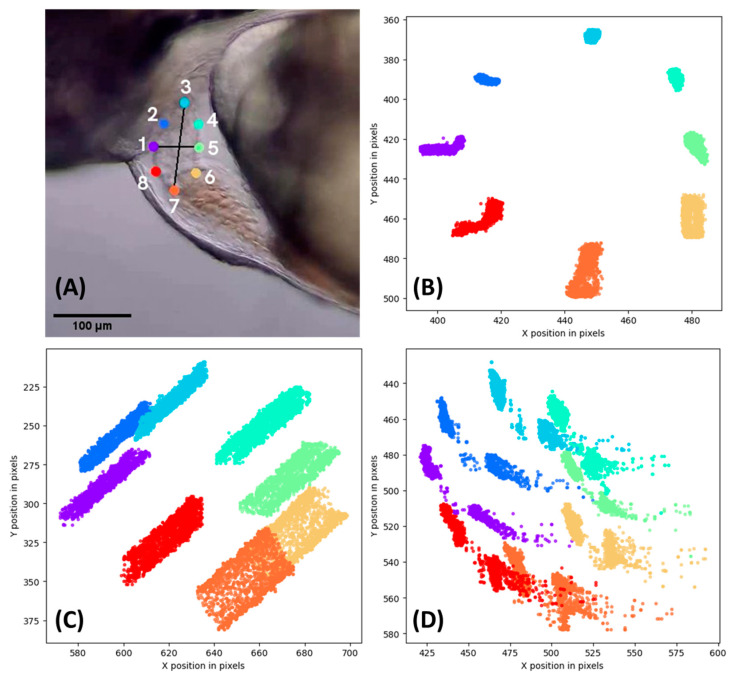
The representative image plots of the ventricle chamber labeled by DLC. (**A**) Eight different circle points label the ventricle of zebrafish larvae. (**B**) Plot created from normal placement without interference. (**C**) Plot created from a microscope slide that was moving slightly. (**D**) Plot created from the zebrafish heart that suddenly moved.

**Figure 4 biology-11-01243-f004:**
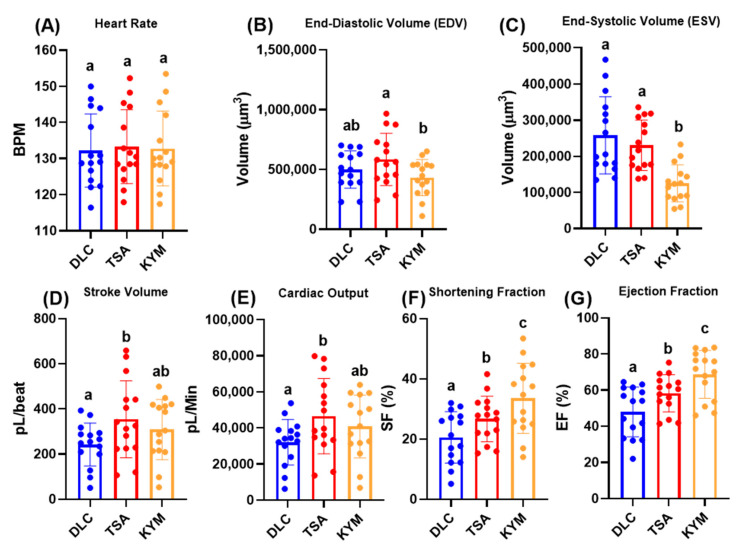
Cardiac physiology parameter comparison between DLC, ImageJ TSA, and Kymograph methods in zebrafish. Results for heart rate (**A**), end-diastolic volume (**B**), end-systolic volume (**C**), normalized stroke volume (**D**), cardiac output (**E**), shortening fraction (**F**), and ejection fraction (**G**), from three methods were compared statistically by RM one-way ANOVA followed by Tukey’s multiple comparisons test. Data are expressed as mean ± SD and significant differences (*p* < 0.05) are indicated by lower case a, b, and c (*n* = 15).

**Figure 5 biology-11-01243-f005:**
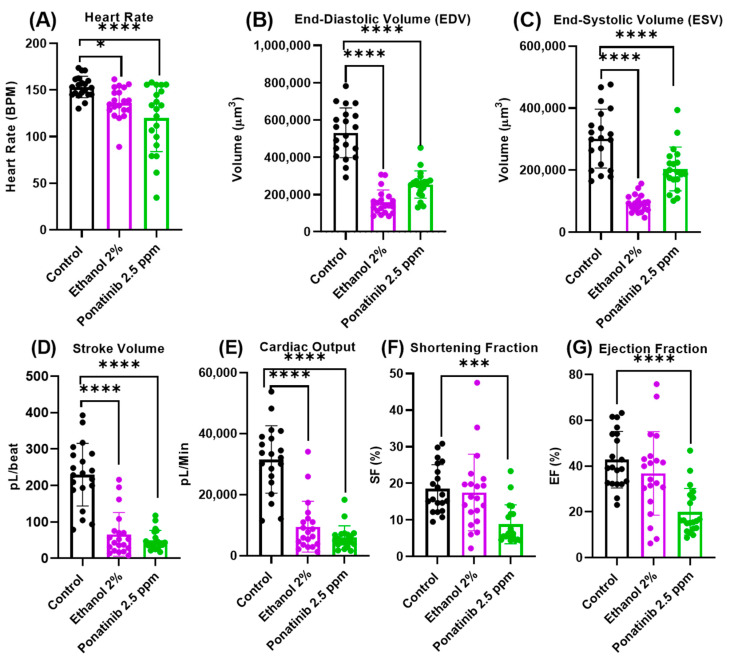
Cardiac physiology parameter comparison between control, ethanol 2%, and ponatinib 2.5 ppm analyzed by DLC method. Heart rate (**A**), end-diastolic volume (**B**), end-systolic volume (**C**), normalized stroke volume (**D**), cardiac output (**E**), shortening fraction (**F**), and ejection fraction (**G**). Results from control and chemical treatment were statistically analyzed by ordinary one-way ANOVA followed by Fisher’s LSD test as post hoc multiple comparison test. Data are expressed as mean ± SD and statistical difference is indicated by * *p* < 0.05; *** *p* < 0.001; and **** *p* < 0.0001 (*n* = 20).

**Figure 6 biology-11-01243-f006:**
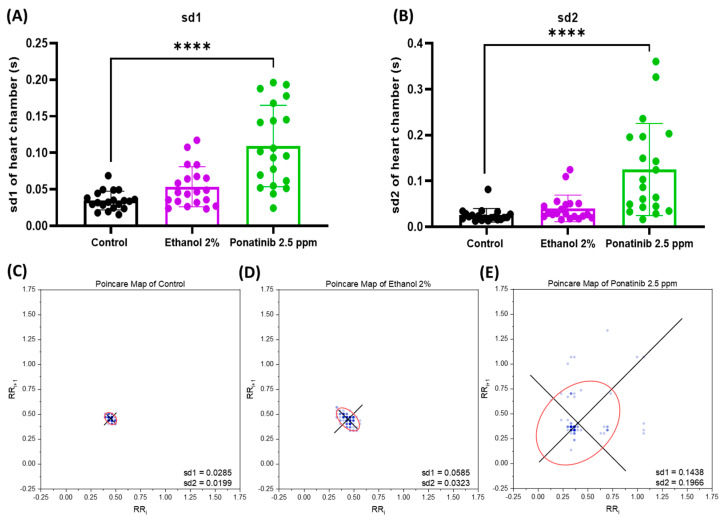
Heart rate variability evaluation in zebrafish embryos after exposure to ethanol 2% and ponatinib 2.5 ppm. Data analyses were conducted based on the standard deviation 1 (sd1) (**A**) and standard deviation 2 (sd2) (**B**) of the heart chamber generated by Poincaré Plot (**C**–**E**). The sd1 and sd2 results were statistically analyzed by Kruskal–Wallis test, followed by the uncorrected Dunn’s test. Data are expressed as mean ± SD, and statistical difference is indicated by **** *p* < 0.0001 (*n* = 20).

**Table 1 biology-11-01243-t001:** Summary of previous detection methods and endpoints for cardiac physiology assessment in zebrafish. Abbreviations: dpf: days post-fertilization; hpf: hours post-fertilization.

Species	Detection Method	Detection Endpoints	Literatures
*Danio rerio* (36–120 hpf)	Detection algorithms written in Matlab based on changes in pixel intensity and color segmentation	Heartbeats and heart rate irregularity	Pylatiuk et al., 2014 [7]
*Danio rerio* (3 dpf)	Using green fluorescent protein-expressing zebrafish Tg(cmlc2: GFP) to automate the myocardial phenotype screening	Number of heart contraction times based on subsite pixel intensities changes	Burns et al., 2005 [8]
*Danio rerio* (72 hpf)	Heart rate was calculated by the software “DanioScope” using a Noldus DanioVision system	Heart rate in beats per minute (BPM) based on video assessment of inter-beat intervals	Zhong et al., 2021 [9]
*Danio rerio*	Kymograph plugin in ImageJ	Heartbeat regularity, stroke volume, ejection fraction, shortening fraction, and cardiac output	Kurnia et al., 2021 [4]
*Danio rerio*	ImageJ based on the dynamic pixel changes method	Atrium rhythm and heartbeat frequency	Santoso et al., 2019 [10]

**Table 2 biology-11-01243-t002:** Summary of deep learning-based methods and endpoints used to conduct cardiac physiology assessment in fish.

Species	Deep Learning Method	Detection Endpoints	Literatures
*Danio rerio* (3 dpf)	Automatic assessment of cardiovascular function based on a U-net deep learning model	Shortening fraction and ejection fraction of masked ventricles	Naderi et al., 2021 [15]
*Danio rerio* (embryonic)	A stand-alone software that uses C# language with the.NET Framework 4.5.2	Shortening fraction based on two pairs of marking points from the diastolic and systolic heart edges of the ventricles	Nasrat et al., 2016 [16]
*Danio rerio* (48 to 96 hpf)	Automatic detection of the heart region by using an intelligent robotic microscope	Heart-region detection based on the intensity and difference images which can be used to distinguish the heart dysfunction	Spomer et al., 2012 [17]
*Danio rerio*	OpenCV-based approach	Heart rate and heartbeat regularity	Farhan et al., 2021 [18]
*Danio rerio*	Cardiac Functional Imaging Network (CFIN)	Shorteing fraction, ejection fraction, heart rate, and cardiac output	Akerberg et al., 2019 [19]
*Danio rerio*	Zebrafish Heart Rate Automatic Method (Z-HRAM)	Heartbeat detection based on zebrafish body expansion and contraction movements	Xing et al., 2018 [20]
*Danio rerio* (2–3 dpf)	DeepLabCut^TM^ (DLC) using ResNet-152	Calculation of volume change, heart rate, stroke volume, ejection fraction, shortening fraction, cardiac output, and heartbeat regularity based on 8-point labeling of heart edges for short and long axis lengths	In this study

## Data Availability

The data presented in this study are available on request from the corresponding author.

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
