# Peer review of "Using DeepLabCut as a Real-Time and Markerless Tool for Cardiac Physiology Assessment in Zebrafish"

_biology, 2022, doi:10.3390/biology11081243_

Round 1
Reviewer 1 Report
This is a well-written manuscript in which the authors employed the DteepLabCut (DLC) tool to evaluate the ventricle morphology in zebrafish and compared the method with the one using the ImageJ plugin. The study showed DLC good potential as an additional tool in the studies of cardiac physiology in fish larvae. Specific comments were provided below on the areas that could be improved before it is published:
1. At the beginning of the study, 20 frames from each video were extracted. The authors are encouraged to clarify how they determined the number of frames and which frames were selected, given that there are 1800 (30fpX60s) frames in the video.
2. Scale bars were missing in both images and movies. Figure 3A: It would be better to number the eight points to outline the ventricle so the reader would know where points 1, 5, 3, and 7 were used to define the long and short axis. Also, the description of “heart1, heart2, heart3…” in 5.3. is confusing, which could be simplified as “1,2,3,….” Figure 3B-D: scales in the y-axis were missing, and the color bars on the right side of the chats can be removed.
3. Figure 4E-G: the author should describe how they get the numbers of pL/beat, pL/Min, SF(%), and EF(%), and explain the physiological outcomes.
4. Figure 4E-G: The authors claimed that DLC provided more precise results than the TSA method, which gave overestimated values. However, they did not provide the actual values that can be normalized to. How do they know which method provides more precise results? The authors should address this issue.
5. The sentence in the first paragraph of 2.2 (“ Results from the zebrafish……. Figure 2A”) is hard to understand. Please rephrase it.
6. Please clarify the “noise potential “ in the middle of 3.2. section “In addition, ROI selection is also limited due to the possible noise potential in certain heart regions”
Author Response
Comments and Suggestions for Authors
This is a well-written manuscript in which the authors employed the DteepLabCut (DLC) tool to evaluate the ventricle morphology in zebrafish and compared the method with the one using the ImageJ plugin. The study showed DLC good potential as an additional tool in the studies of cardiac physiology in fish larvae. Specific comments were provided below on the areas that could be improved before it is published:
- At the beginning of the study, 20 frames from each video were extracted. The authors are encouraged to clarify how they determined the number of frames and which frames were selected, given that there are 1800 (30fpX60s) frames in the video.
Thank you for pointing out this matter. The selection number of 20 frames was based on the default setting and the frames were automatically selected based on k-means algorithm. In the DLC, the provided function for selecting frames from the videos in a temporally uniformly distributed way (uniform), either by clustering based on visual appearance (k-means), or by manual selection [1]. K-means was selected to further prevent intervention from human error. People might select the frames that displayed similar images or shapes of heart, thus, not adding any value of variables for the training step. K-means is one of the most popular, given its simplicity and computational speed, an important feature when there are large amounts of images to be processed, and capacity to deal with a large number of variables The k-means method clusters the frames, then this function downsamples the video and clusters the frames [2]. Frames from different clusters are then selected. This procedure ensures that the frames look different and is generally preferable. The explanation regarding the extracted frames by K-means has been added to the revised manuscript (line 130-132). In addition, the extraction of 20 frames from each video has been proven enough for pose estimation, identification, and tracking which has been reported by previous studies [3,4]. The developer also recommends that it’s enough to have overall total ~200 frames for train model [5]. We used overall 40 videos for training which means we have total 800 frames which is sufficient for the training process.
- Nath, T., Mathis, A., Chen, A. C., Patel, A., Bethge, M., & Mathis, M. W. (2019). Using DeepLabCut for 3D markerless pose estimation across species and behaviors. Nature protocols, 14(7), 2152-2176.
- Jardim, S., António, J., & Mora, C. (2022). Graphical Image Region Extraction with K-Means Clustering and Watershed. Journal of Imaging, 8(6), 163.
- Lauer, J., Zhou, M., Ye, S., Menegas, W., Schneider, S., Nath, T., ... & Mathis, A. (2022). Multi-animal pose estimation, identification and tracking with DeepLabCut. Nature Methods, 19(4), 496-504.
- Haberfehlner, H., van de Ven, S. S., van der Burg, S., Aleo, I., Bonouvrié, L. A., Harlaar, J., ... & van der Krogt, M. M. (2022). Using DeepLabCut for tracking body landmarks in videos of children with dyskinetic cerebral palsy: a working methodology. medRxiv.
- Mathis, A., Mamidanna, P., Cury, K. M., Abe, T., Murthy, V. N., Mathis, M. W., & Bethge, M. (2018). DeepLabCut: markerless pose estimation of user-defined body parts with deep learning. Nature neuroscience, 21(9), 1281-1289.
2. Scale bars were missing in both images and movies. Figure 3A: It would be better to number the eight points to outline the ventricle so the reader would know where points 1, 5, 3, and 7 were used to define the long and short axis. Also, the description of “heart1, heart2, heart3…” in 5.3. is confusing, which could be simplified as “1,2,3,….” Figure 3B-D: scales in the y-axis were missing, and the color bars on the right side of the chats can be removed.
Thank you for the detailed check and valuable suggestions. The scale bars have been added in all images and videos contained in the manuscript. As your suggestion, the eight number points (1-8) have been added to the Figure 3A. The description name heart1, heart2, heart3, etc have also been simplified to 1, 2, 3, etc. In addition, the Figure 3B-D have also been renewed with y-axis included in the images and the color bars have been removed.
- Figure 4E-G: the author should describe how they get the numbers of pL/beat, pL/Min, SF(%), and EF(%), and explain the physiological outcomes.
Thank you for suggestions. The formulations and physiological outcomes of stroke volume, cardiac output, shortening fraction, and ejection fraction have been added in the materials and methods section 2.4.
- Figure 4E-G: The authors claimed that DLC provided more precise results than the TSA method, which gave overestimated values. However, they did not provide the actual values that can be normalized to. How do they know which method provides more precise results? The authors should address this issue.
Thank you for your valuable comment and suggestion. The authors agree with the reviewer since there was no actual values that can be normalized between the DLC and TSA, thus, it is uncertain which method provides more precise results. The current DLC method analyzed the cardiac parameter based on the changes in the heart chamber size, meanwhile the TSA used dynamic pixel intensity changes. These two methods used different approaches, however based on the simple measurement, such as heartbeat similar results were retrieved. Regarding to the statement that DLC provide more results than the TSA method, the authors also find this sentence is problematic and exaggerated. Thus, the authors have removed the sentence. However, despite of the absence of the actual values comparison that can be normalized, DLC still has additional value with real-time monitoring provided by the labeled video and xy positions from each frame. In contrary to the TSA that manually measured the heart’s diameter at the diastolic stage (heart relaxation) and systolic stage (heart contraction) by drawing straight line on single frame as representative in ImageJ.
- The sentence in the first paragraph of 2.2 (“ Results from the zebrafish……. Figure 2A”) is hard to understand. Please rephrase it.
Thank you for your suggestion. The sentence has been rephrased to “The evaluation results showed that ResNet-101 had the lowest accuracy. It’s proven from the results of training and test error rates which were statistically higher than ResNet 50 & 152 (Figure 2A)”.
- Please clarify the “noise potential “ in the middle of 3.2. section “In addition, ROI selection is also limited due to the possible noise potential in certain heart regions”
Thank you for your suggestion. The noise potential in here means the image noise that might be occurred by some random variation of brightness, which can be caused by unwanted signal, effect of sensor size, level of illumination, temperature, etc. This will cause some regions to have darker or brighter pixels. High levels of noise are almost always undesirable, which can also influence the user in determining the ROI.
Reviewer 2 Report
In their manuscript “Using DeepLabCut as Real-Time and Markerless Tool for Cardiac Physiology Assessment in Zebrafish”, the authors describe how they applied a deep learning tool originally developed for animal pose determination to automatically determine heart beat and other parameters of cardiac function in zebrafish embryos. ResNet evaluation networks were used to evaluate error rates, identifying one network (ResNet-152) as performing best. DeepLabCut (DLC) was able to label the ventricle even when the embryo was sliding under the objective or made spontaneous sudden movements. The authors used the acquired data to compare the performance of DLC to another previously used method (ImageJ Time Series Analysis, TSA) and found that for some parameters, DLC provided lower values than TSA, while heart rate and end-systolic volume values were not different between the two methods. They also treated embryos with ethanol and ponatinib to trigger cardiac malformation and malfunction, and DLC was able to detect differences in the treated embryos, both in terms of cardiac physiology parameters and heart rate variability.
While the new application of the deep learning tool to zebrafish embryo heart physiology appears promising, the manuscript still lacks a more rigorous benchmarking evaluation.
Thus, the authors estimate that the TSA method generates “14-22% higher outcomes than the DLC method” for some cardiac parameters and claim that, therefore, “a deep learning-based method can provide better and more precise results that the ImageJ method, which could give an overestimated value”. Couldn’t one also interpret these results in the opposite way, namely that TSA is closer to the “real” values and that DLC underestimates the values? Is there additional information available which allows one to decide between these two possibilities?
Furthermore, the tool should be benchmarked against at least one other tool that equally relies on deep learning approaches, to allow for a comparison to a similar methodological approach.
Minor points
Please provide more information on how the cardiac physiology parameters (EDV, ESV, stroke volume, cardiac output, shortening fraction, ejection fraction, sd1 and sd2 of heart rate variability) are defined and calculated. Which of these parameters are independent from each other?
The first and part of the second paragraph of the discussion (page 8) provide a lot of information on the design of the method that would be better placed at the beginning of the results section. Similarly, the description of how the DLC’s effectiveness was calculated (page9/10) should be moved to the results section.
Is there a specific reason for using 50,00 to 500,000 iterations? What is meant by “we validated with one number of shuffles, which resulted in the test and train error (X and Y pixels)”?
Figure A1: replace “labeled” with “labelling”.
Author Response
Comments and Suggestions for Authors
In their manuscript “Using DeepLabCut as Real-Time and Markerless Tool for Cardiac Physiology Assessment in Zebrafish”, the authors describe how they applied a deep learning tool originally developed for animal pose determination to automatically determine heart beat and other parameters of cardiac function in zebrafish embryos. ResNet evaluation networks were used to evaluate error rates, identifying one network (ResNet-152) as performing best. DeepLabCut (DLC) was able to label the ventricle even when the embryo was sliding under the objective or made spontaneous sudden movements. The authors used the acquired data to compare the performance of DLC to another previously used method (ImageJ Time Series Analysis, TSA) and found that for some parameters, DLC provided lower values than TSA, while heart rate and end-systolic volume values were not different between the two methods. They also treated embryos with ethanol and ponatinib to trigger cardiac malformation and malfunction, and DLC was able to detect differences in the treated embryos, both in terms of cardiac physiology parameters and heart rate variability.
While the new application of the deep learning tool to zebrafish embryo heart physiology appears promising, the manuscript still lacks a more rigorous benchmarking evaluation. Thus, the authors estimate that the TSA method generates “14-22% higher outcomes than the DLC method” for some cardiac parameters and claim that, therefore, “a deep learning-based method can provide better and more precise results that the ImageJ method, which could give an overestimated value”. Couldn’t one also interpret these results in the opposite way, namely that TSA is closer to the “real” values and that DLC underestimates the values? Is there additional information available which allows one to decide between these two possibilities?
The authors thank the reviewer for the questions. The authors agree with the reviewer since this statement may be misinterpreted and it is uncertain which method provides more precise results. Regarding to the statement “a deep learning-based method can provide better and more precise results that the ImageJ method, which could give an overestimated value”, the authors also found it might be exaggerated without solid values comparison that can be normalized. Thus, the authors have removed the sentence. However, several things to note, the current DLC method analyzed the cardiac parameter with different approach based on the changes in the heart chamber size, meanwhile the TSA used dynamic pixel intensity changes to measure the cardiac rhythm. To determine the other cardiac parameters in TSA method, manual measurement is required. In TSA method, we manually measured the heart’s diameter at the diastolic stage (heart relaxation) and systolic stage (heart contraction) by drawing straight line on single frame as representative in ImageJ [1]. In contrary, DLC detect the heart’s diameter with real-time labeling and provided with xy positions from each frame. This might become the advantage of the newly developed method DLC and address the limitation in TSA method, because we can assume that real-time labeling in the whole frames of video is much better than just few representative frames. Additional information regarding this matter has been added to the discussion section. The conclusion has also been revised to “This study also revealed that the DLC method displayed identical results in term of cardiac rhythm compared to the previous published TSA and KYM method. However, in some cardiac parameters, different results were obtained which might be due to manual measurement and ROI selection dependency from ImageJ based method. Compared to these previous methods, the DLC has the advantages of real-time labeling in the whole frames of video with fully automation” (line 494-499).
- Sampurna, B.P.; Audira, G.; Juniardi, S.; Lai, Y.-H.; Hsiao, C.-D. A simple imagej-based method to measure cardiac rhythm in zebrafish embryos. Inventions 2018, 3, 21.
Furthermore, the tool should be benchmarked against at least one other tool that equally relies on deep learning approaches, to allow for a comparison to a similar methodological approach.
Thank you so much for your concern and suggestion. However, currently, we do not have another tool in hand that equally relies on a deep learning approach. Other problems, most of other deep-learning tool in literatures did not provide complete source codes and some use different programming language. In addition, it’s not possible to adjusted and optimized other deep-learning tool for training, evaluation, test, and detection our recorded videos in this short time. Instead of that, we use our previous published Kymograph method for another data comparison. The kymograph method reported here can measure cardiac rhythm either by the blood cells within the heart chamber or heart chamber movement (contraction and relaxation) similar with the current DLC approach, providing a more flexible option choice for the users [2]. The Figure 4 has been updated with cardiac parameters result from Kymograph method (line 331-352). In terms of cardiac rhythm, identical results of heart rate were displayed, however in some other cardiac parameters different results were obtained. Each method in this study has measured the heart rate with identical results, which prove that our methods successfully identified and determined the diastolic-systolic stage (heartbeats) either by pixel intensity changes or chamber movement. However, when come to specific measurements, such as EDV and ESV, different results were given. Like TSA, the KYM outcome is mainly affected by the ROI selection. Heart chamber was measured by drawing a line from the heart’s inner part to the outermost area during diastole. The line must go through the ROI where heart movement or blood is present. This problem has been addressed in the discussion section method and regarding which method provides better and more precise results in this study cannot be decided yet since no actual normalized data comparison. This also become the limitation of this study and the authors definitely agree with the reviewer on the importance of the comparison with other deep-learning tools. Hopefully, this matter can be addressed and resolved in the future study.
- Kurnia, K.A.; Saputra, F.; Roldan, M.J.M.; Castillo, A.L.; Huang, J.-C.; Chen, K.H.-C.; Lai, H.-T.; Hsiao, C.-D. Measurement of Multiple Cardiac Performance Endpoints in Daphnia and Zebrafish by Kymograph. Inventions 2021, 6, 8.
Minor points
Please provide more information on how the cardiac physiology parameters (EDV, ESV, stroke volume, cardiac output, shortening fraction, ejection fraction, sd1 and sd2 of heart rate variability) are defined and calculated. Which of these parameters are independent from each other?
Thank you so much for your valuable suggestions and questions. The cardiac physiology parameters information and calculation have been added to the manuscript (Materials and Methods section 2.4). Based on the diameter of short and long axis of ventricle chamber, the EDV and ESV can be calculated. Then, the heart rate was determined based on the peaks of diastolic-systolic phase in 1 minute. After EDV, ESV, and heart rate were obtained, other parameters such as stroke volume, cardiac output, and ejection fraction can be measured. One parameter might be different from others is shortening fraction because it’s measurement only based on the short axis diameter. Meanwhile, the other parameters need volume measurements. The heart rate variability information and measurement have also been added to the manuscript (Materials and Methods section 2.7).
The first and part of the second paragraph of the discussion (page 8) provide a lot of information on the design of the method that would be better placed at the beginning of the results section. Similarly, the description of how the DLC’s effectiveness was calculated (page9/10) should be moved to the results section.
Thank you so much for the suggestions. The first paragraph of the discussion section has been moved to the beginning of results section (line 236-249), while the second paragraph of the previous discussion section has been moved results section 3.2 (line 288-296) to as your suggestion. In addition, the description of DLC’s effectiveness has also been moved to line 302-310.
Is there a specific reason for using 50,00 to 500,000 iterations? What is meant by “we validated with one number of shuffles, which resulted in the test and train error (X and Y pixels)”?
Thank you for your questions. The iterations were chosen based on the minimum recommended number iterations which is 50,000 and the maximum: 500,000 that was reported enough for convergence in the presented cases by the developer [3]. And in this study, the 500,000 iterations were proven enough to detect and label the heart chamber. We apologize for the mistake; the intended meaning is ten number of shuffles (from 50,000; 100,000; 150,000; ….; until 500,000), which resulted in the evaluation network of test and train error rate (based on X and Y pixels position) in the Figure 2.
- Mathis, A.; Mamidanna, P.; Cury, K.M.; Abe, T.; Murthy, V.N.; Mathis, M.W.; Bethge, M. DeepLabCut: markerless pose estimation of user-defined body parts with deep learning. Nature neuroscience 2018, 21, 1281-1289.
Figure A1: replace “labeled” with “labelling”.
Thank you so much for the correction. The word “labeled” in Figure A1 has been replaced with “labelling”.
Round 2
Reviewer 2 Report
The authors have addressed the issues I raised appropriately. I have but a few additional minor suggestions that the authors might consider for the final manuscript version:
Simple summary, line 23: “…compared with the previous published method...” should read “compared with the previous published methods…”. Same for the abstract, line 40: methods. Similar minor grammatical errors are still present here and there in the manuscript and should be corrected prior to publication.
Upon moving the entire former second paragraph of the discussion into the results section, there are now some redundancies in lines 285-289 that could be merged.
Author Response
Comments and Suggestions for Authors
The authors have addressed the issues I raised appropriately. I have but a few additional minor suggestions that the authors might consider for the final manuscript version:
The authors highly appreciate the reviewer for taking necessary time and effort to review this manuscript. We would like to take this opportunity to thank you for the insight and expertise that you contributed towards reviewing the article, which helped us in improving the quality of this manuscript.
Simple summary, line 23: “…compared with the previous published method...” should read “compared with the previous published methods…”. Same for the abstract, line 40: methods. Similar minor grammatical errors are still present here and there in the manuscript and should be corrected prior to publication.
Thank you so much for your correction. The grammatical check has been done and some sentences in the manuscript has been revised and corrected.
Upon moving the entire former second paragraph of the discussion into the results section, there are now some redundancies in lines 285-289 that could be merged.
Thank you so much for your detailed check. The authors agree with the reviewer and also found the redundancies, therefore, the sentences in line 285-289 have been removed.